# Vibration Energy Coupling Behavior of Rolling Mills under Double Disturbance Conditions

**Lidong Wang [1], Shen Wang [2], Xingdou Jia [1], Xiaoling Wang [1] and Xiaoqiang Yan [1,\*]**

[1] School of Mechanical Engineering, University of Science and Technology Beijing, Beijing 100083, China
[2] Department of Mechanical Engineering and Mechanics, Lehigh University, Bethlehem, PA 18015, USA
\* Correspondence: yanxq@ustb.edu.cn; Tel.: +86-186-0026-0898

**Abstract:** The operation of the world's first multimode continuous casting and rolling F3 (3rd finishing mill stand) finishing mill was hampered by frequent vibrations. Mill vibrations were found to be caused by the transmission and coupling of vibration energy flow. In this study, an overall finite element model of the F3 stand is established based on the structural sound intensity method and harmonic response analysis method, and then, the intrinsic energy flow modes and energy flow harmonic response of the F3 stand are obtained. Further, the effects of the steady-state rolling force variation, preload torque variation, rolling force fluctuation, torque fluctuation, and its phase angle difference on the vibration energy flow of the mill are analyzed. Finally, the effects of the mill damping ratio, strip width, and strip modulus on the vibration energy flow under double dynamic load are discussed to reveal the inherent characteristics of the mill vibration energy flow. The results show that the vibration energy flow of the mill increases with the increase of strip modulus, rolling force, and moment fluctuation; the phase angle difference of rolling moment shows a "V" trend change on the vibration energy flow; the change of strip width has a greater effect on the vibration energy flow of the vertical system; and for the damping ratio of 0.01–0.1, the reduction of the vibration energy flow at all excitation frequencies is obvious.

**Keywords:** rolling mill vibration; energy flow mode; coupling system; energy flow harmonic response





## 1. Introduction

Mill vibration has been extensively studied by researchers worldwide. Specifically, researchers have investigated the vibration mechanism and vibration control of rolling mills from three aspects: structural parameters, process parameters, and nonlinear coupled dynamics models.

In terms of structural parameter changes, Liu et al. [1] studied the effect of nonlinear stiffness variation of hydraulic cylinders on the stability of dip vibration in the automatic control system of a roll seam. Zhang et al. [2] established a nonlinear horizontal–vertical coupling model of a mill and studied the effects of the structural clearance and magnitude of external excitation fluctuations on the stability of the system. Researchers have investigated the dynamic behavior of work roll bearings in finishing and roughing mills under various failure conditions [3,4]. Other researchers investigated the nonlinear torsional vibration caused by the mill jointing shaft and effectively controlled the torsional vibration by using an active control method [5–7]. Xu et al. [8] analyzed the effect of the structural clearance of the main drive system on vibration stability based on the multibody dynamics theory.

In terms of process parameter changes, Fan et al. [9] investigated the effect of lubrication factors on system vibration at the rolling interface under different process conditions. Peng et al. [10] studied and established a hysteretic nonlinear rolling force model for a four-roller mill to analyze the effect of different rolling process parameters on the vibration amplitude and frequency characteristics. Zhang et al. [11] established a digital twin vibration monitoring model for four-roll mills and found that, the greater the rolling force fluctuation, the more pronounced the vibration of the mill roll system.

In terms of nonlinear coupled dynamics models, Papini et al. [12] proposed various nonlinear rolling models based on Qrowan's theory and performed vibration simulation analyses. Liu et al. [13] applied Hill's rolling force theory to establish a roll system-interface coupling dynamic rolling force model and found that optimizing the time lag parameters can effectively suppress mill vibration. Ha-Nui et al. [14] discussed the statistical and monitoring of mill chatter response using independent analysis methods. In the literature [15,16], it is considered that the coupling relationship between the mill structure and the rolling process is due to the vertical-torsional-horizontal coupling, and the changes in the parameters of the rolling process cause changes in the critical boundaries of the mill vertical vibration mode, horizontal vibration mode, and torsional vibration mode.

Most of the abovementioned studies used only a local analysis to study the resonance problem, and they failed to effectively solve long-term mill vibrations. Notably, the vibration of the entire mill stand is essentially a dynamic process of transmission and mutual coupling in the form of vibration energy waves. However, few studies have investigated the vibration energy transfer and coupling characteristics of a hot strip mill, and the vibration energy transfer law and coupling behavior in the stand remain unclear. Therefore, it is reasonable to clarify mill vibrations from the perspective of vibration energy.

In this article, a dynamics model of the F3 stand of the finishing unit of the MCCR hot strip mill was established based on the structural sound intensity method of the combined shell-beam-cylinder unit, and the sensitivity frequency of the mill vertical-torsion coupling vibration was calculated using the ANSYS software harmonic response and verified through field experiments. Furthermore, the energy flow mode and energy flow harmonic response cloud diagram of the mill were analyzed, and the energy flow harmonic response vector diagram of the mill was obtained using MATLAB software. Finally, the effects of steady-state rolling force variation, preload torque variation, rolling force fluctuation, torque fluctuation, and phase angle difference between the two as well as those of the mill damping ratio, bandwidth, and strip modulus on the energy flow were studied.

In this study, the mill vibration energy flow model and vibration energy flow harmonic response of the F3 stand of a multimode continuous casting and rolling plant (MCCR) finishing mill are investigated under double disturbance conditions through numerical calculations. Then, the factors affecting the maximum value of the mill vibration energy flow are analyzed further, and corresponding conclusions are drawn. Section 2 presents the principles of numerical calculations of the F3 stand (shell-beam-cylinder combination) model based on the structural sound intensity method. Section 3 discusses the amplitude—frequency characteristics of the mill drive-roll-vertical system. Section 4 presents the results of the mill vibration energy flow calculations and discusses its influencing factors. Finally, Section 5 presents the conclusions of the study.

## 2. Shell-Beam-Cylinder Combination Model Calculations Using Sound Intensity Method

The essence of mill vibration is the energy transfer process. The flow of vibrational energy transferred to a region of the structure can be described by the structural sound intensity method, which combines force and velocity at any point in an elastic structure to characterize the flow of energy in a vibrating structure. For steady-state vibration of a complex structure, this structural intensity can be defined as [17]

$$\prod_k(\omega) = -\frac{1}{2}\sum \sigma_{kl}(\omega)v_l^*(\omega) = I_k(\omega) + jJ_k(\omega) \quad k,l = 1,2,3 \tag{1}$$

where $\omega$ is the vibration frequency; $\sigma_{kl}(\omega)$, the stress tensor components; $v_l^*(\omega)$, the conjugation of velocity vector components; $I_k(\omega)$, the active strength; and $J_k(\omega)$, the reactive intensity.

Structural intensity streamlines show the flow as a smooth curve parallel to the velocity field, with the relative spacing of the lines indicating the velocity of the streamlines. The

steady-state structural sound intensity flowline can be defined as in [16], where r is the energy flow particle position. The mill support is a complex assembly of coupled beam elements, shell elements, and cylinder elements, as shown in Figure 1. The mill model is simplified by removing the chamfers of each part without affecting the quality of the model. The mill model is meshed with a hexahedral mesh.

$$\mathbf{dr} \times \mathbf{I}(\mathbf{r}, t) = \begin{vmatrix} \mathbf{i} & \mathbf{j} & \mathbf{k} \\ I_x & I_y & I_z \\ \mathrm{d}x & \mathrm{d}y & \mathrm{d}z \end{vmatrix} = 0 \tag{2}$$

A Fourier transform can be used to obtain the expression of the vibration power flow in the frequency domain for the cylinder elements, shell elements, and beam elements [18,19]. As shown in Figure 1, shell elements are used to simulate thin-walled parts such as universal joint housings and upper and lower crossbeams of the mill; beam elements are used to simulate intermediate shafts, work rolls, etc.; and cylindrical elements are used to simulate support rolls, main motors, etc.

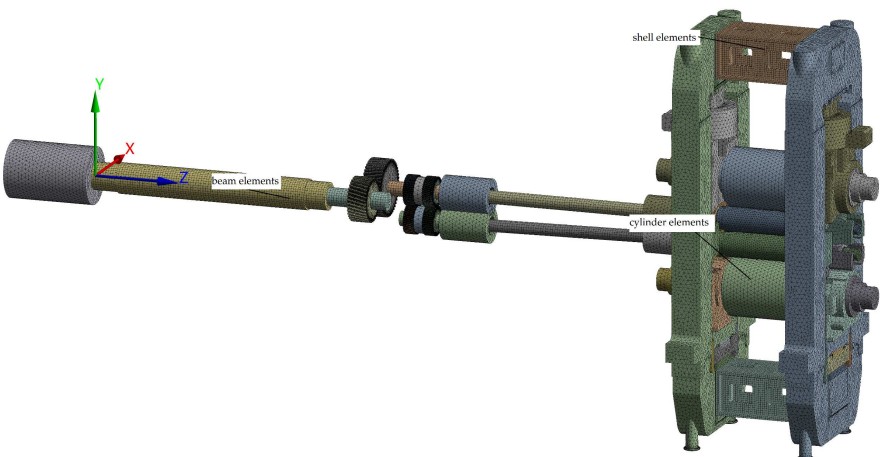

**Figure 1.** Finite element model of shell-beam-cylinder coupled system.

- By Fourier transform, the structural sound intensity of the cylindrical elements in the frequency domain can be expressed as follows;

$$\begin{cases} I_x^{cylinder} = -(\omega/2)\mathrm{Im}\left[\sigma_{xx}\dot{u}_x + \sigma_{xy}\dot{u}_y + \sigma_{xz}\dot{u}_z\right] \\ I_y^{cylinder} = -(\omega/2)\mathrm{Im}\left[\sigma_{yx}\dot{u}_x + \sigma_{yy}\dot{u}_y + \sigma_{yz}\dot{u}_z\right] \\ I_z^{cylinder} = -(\omega/2)\mathrm{Im}\left[\sigma_{zx}\dot{u}_x + \sigma_{zy}\dot{u}_y + \sigma_{zz}\dot{u}_z\right] \end{cases} \tag{3}$$

- By Fourier transform, the structural sound intensity of the shell elements in the frequency domain can be expressed as follows;

$$\begin{cases} I_x^{shell} = -(\omega/2)\mathrm{Im}\left[N_x\dot{u} + N_{xy}\dot{v} + Q_x\dot{w} + M_x\dot{\theta}_y - M_{xy}\dot{\theta}_x\right] \\ I_y^{shell} = -(\omega/2)\mathrm{Im}\left[N_v\dot{v} + N_{yx}\dot{u} + Q_y\dot{w} - M_y\dot{\theta}_x + M_{yx}\dot{\theta}_y\right] \end{cases} \tag{4}$$

- By Fourier transform, the structural sound intensity of the beam elements in the frequency domain can be expressed as follows;

$$I_x^{beam} = -(\omega/2)\mathrm{Im}\left(N_x\dot{u}_x + Q_y\dot{u}_y + Q_z\dot{u}_z + T\dot{\theta}_x + M_y\dot{\theta}_y + M_z\dot{\theta}_z\right) \tag{5}$$

where $P_x$, $P_y$, and $P_z$ are the power flows in the three directions of the unit.

## 3. Amplitude and Frequency Characteristics of Rolling Mill Drive-Roll-Vertical System

### 3.1. Finite Element Modeling of Coupled Structure of Mill

The world's first integrated MCCR was used to simulate the F3 stand of the finishing mill, assuming that the whole stand is composed of a homogeneous pure elastic material. The solid model is shown in Figure 2.

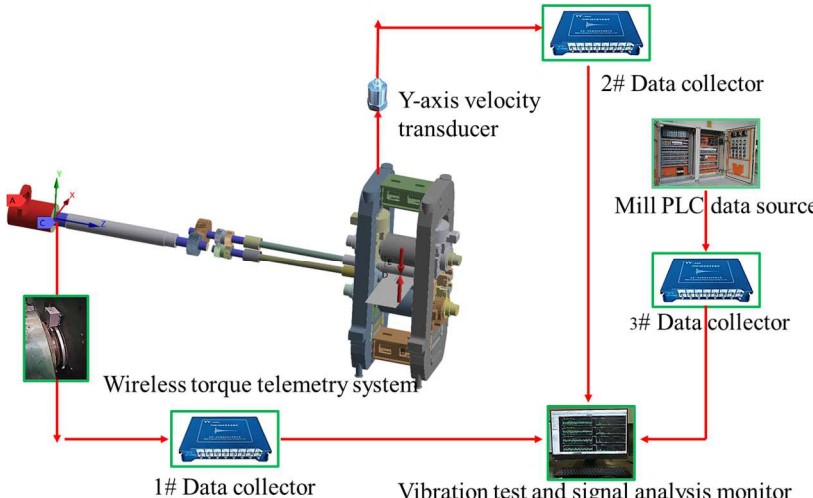

**Figure 2.** F3 rack finite element analysis and online monitoring.

The structural parameters of the F3 model shown in Figure 2 are listed in Table 1.

**Table 1.** Structural parameters.

| Parameter | Specification |
| --- | --- |
| Backup roll diameter | 1450 mm |
| Motor power | 10,000 kW |
| Number of motor stages | 6 |
| Motor speed | 230–560 rpm |
| Work roll length | 4122 mm |
| Constant structural damping factor | 0.018 |
| Work roll diameter | 640 mm |
| Motor rotor moment of inertia | 6217 kg·m$^2$ |
| Reduction ratio | 1:1.3 |
| Reducer moment of inertia | 1370 kg·m$^2$ |
| Number of teeth in reducer | 36/47 |
| Gear seat number of teeth | 47 |
| Gear seat moment of inertia | 2350 kg·m$^2$ |
| Poisson's ratio | 0.3 |
| Young's modulus | $2.1 \times 10^{11}$ GPa |
| Density | 7800 kg·m$^{-3}$ |
| Plate rolls | SMCC |
| Strip width | 1280 mm |
| Rolling thickness | 1.5 mm |

In keeping with the actual on-site operation of the mill, a uniform torsional vibration load was applied to the main motor, and a pair of linear pressures of equal size and opposite directions were loaded on the upper and lower work rolls. During long-term field measurements, all inherent frequencies of the mill stand were not excited under different external excitations, and the general vibration frequency of the hot strip mill was within 200 Hz; therefore, the excitation frequency was taken as 0–200 Hz. Rolling mill stand modeling considerations are as follows: (1) Support roll and work roll bearing seat has a guide flange embedded in the guide groove of the plaque; the bearing seat can slide up

and down, with no axial movement; (2) the roll shoulder and bearing seat contact surface is set to not separate and the gear system contact mode is set to not separate; (3) for the support roll and work roll contact, a friction coefficient is set of 0.01; (4) the four-foot planes of the mill stand to apply fixed constraints; (5) for the bearings in the solver control, pure compression support constraints are added, and weak spring is set to open state; (6) and for the mill work roll, intermediate roll, and support roll, the material is steel; the other parts of the frame and the bearing seat material is cast steel.

The torsional vibration measurement of the main drive system is tested using a wireless torque telemetry system that consists of two main components: a rotating ring with an integrated transmitter module and a stationary power stationary ring with a master control unit. The main parameters of the wireless torque telemetry system are shown in Table 2.

**Table 2.** Torsional vibration measurement equipment parameters.

| Parameter | Specification |
|---|---|
| Shaft diameter | 100–1000 mm |
| Spinning speed | <9000 rpm |
| Sampling frequency | 2048 Hz/CH |
| Input range | ±2.5 V |
| Strain accuracy | 0.3% |
| Power consumption | 5 V 100 mA |
| AD accuracy | 24-bit delta-sigma chip |
| Number of channels | 1 |

According to the actual operating condition of the motor at work on site, the torsional vibration load is applied to the motor as follows.

$$\begin{cases} M = M_0 + \Delta M \sin 2\pi f t \\ M_0 = 1.7e + 008N \cdot mm \\ \Delta M = 1.7e + 007N \cdot mm \end{cases} \tag{6}$$

In order to truly simulate the rolling process of the mill, the rolling forces applied between the upper and lower working rolls are as follows.

$$\begin{cases} F = F_0 + \Delta F \sin 2\pi f t \\ F_0 = 1.081 + 004N/mm \\ \Delta F_0 = 1.081 + 003N/mm \end{cases} \tag{7}$$

where $M_0$ is the torque stability value; $\Delta M$, the torque fluctuation amplitude; $\Delta F$, the rolling force fluctuation; $F_0$, the steady-state rolling force; and $f$, the torsional vibration frequency.

As shown in Figure 3, the main drive train section exhibits two resonant response frequencies of 21 Hz and 43 Hz in the harmonic response analysis results and frequent vibrations at 18 Hz (i.e., close to 21 Hz) in the field measurement. Furthermore, the vertical system part shows two resonant response frequencies at 64 Hz and 121 Hz in the simulation and frequent vibrations at 58.5 Hz (i.e., close to 64 Hz) in the field test. After 18 months of online monitoring by our group at the mill site, the main drive system consistently did not exhibit a 43 Hz vibration frequency, while the vertical system did not exhibit a 121 Hz vibration frequency. These phenomena indicate that the mill has multiple inherent frequencies, resonance does not occur at all inherent frequencies, and frequent vibrations occur at only some frequencies.

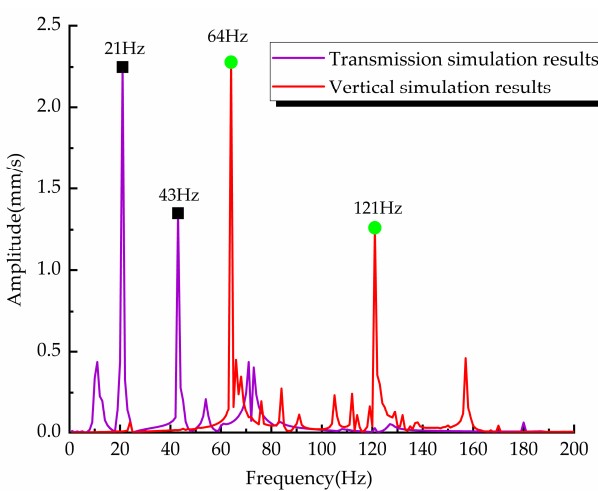

**Figure 3.** Harmonic response simulation results.

## 3.2. Calculated Results of Energy Flow Modal Analysis of Mill

As shown in Figure 4, the vibration of the 6th-order energy flow mode (~21 Hz) is in the form of front-to-back twisting of the transmission part, and that of the 10th-order energy flow mode (~43 Hz) is in the form of up-and-down twisting of the transmission part. However, the vibration of the vertical system in the 6th- and 10th-order energy flow modes is very small. That of the 16th-order energy flow mode (~64 Hz) is in the form of clockwise twisting of the vertical part along the vertical axis, and that of the 24th-order energy flow mode (~121 Hz) is in the form of counterclockwise twisting of the vertical part along the vertical axis. Additionally, the vibration of the main drive in the 16th- and 24th-order energy flow modes is very small.

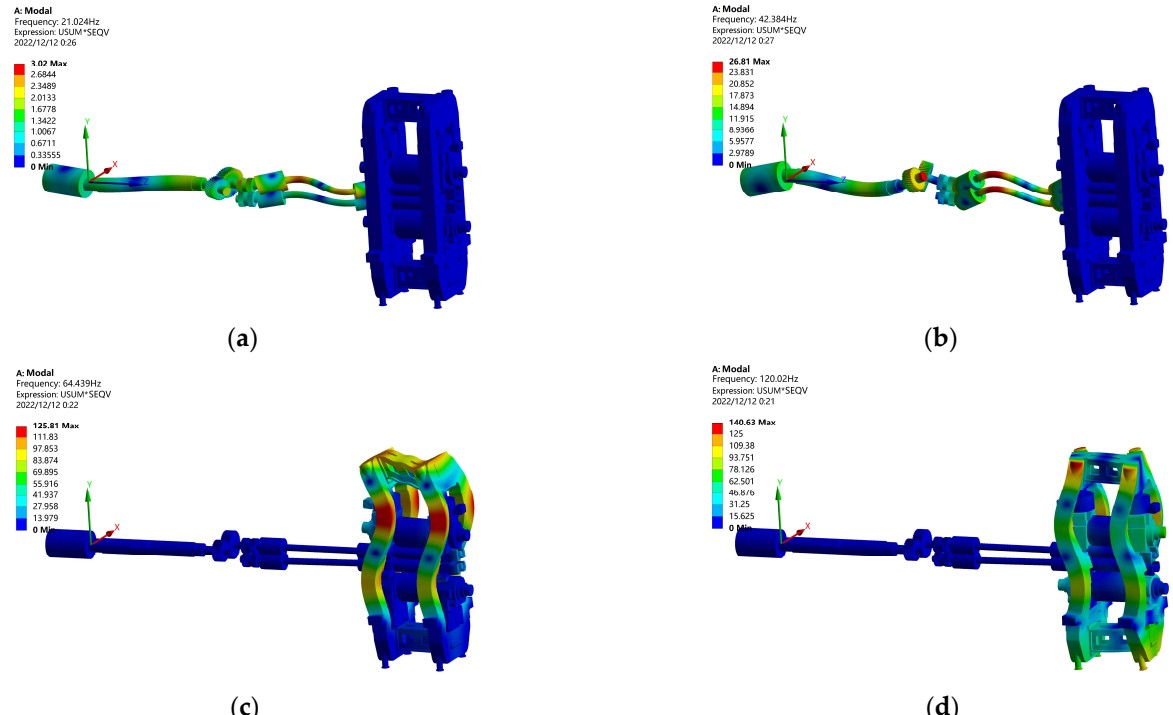

**Figure 4.** Power flow modes: (**a**) 6th-order mode, (**b**) 10th-order mode, (**c**) 16th-order mode, and (**d**) 24th-order mode.

## 4. Mill Vibration Energy Flow Calculation Results and Discussion

### 4.1. Harmonic Response Analysis of Mill Energy Flow

The maximum vibration energy flow of 21 Hz occurs in the lower universal joint drive measurement with an amplitude of 852.07 kW/m² (Figure 5). The first half of the mill universal joint shaft vibration energy is also relatively large, and the mill is the main drive part of the transmission gear vice surface energy concentration. The minimum value of energy flow occurs at the foot of the vertical system.

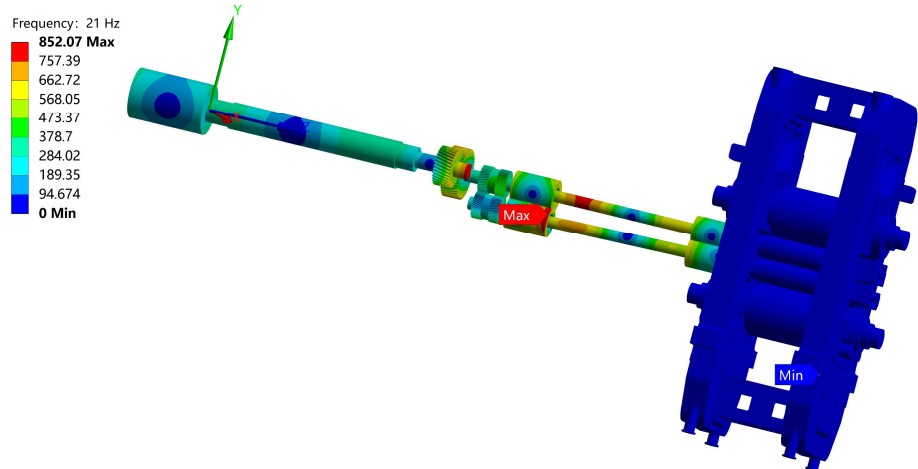

**Figure 5.** Harmonic response cloud for vibration energy flow of 21 Hz.

The energy flow vector on the mill drive system under excitation of 21 Hz decreases gradually from the reducer gear and universal joint shaft drive measurement to the sides of each of its departments (Figure 6). Several energy sinks appear in the main drive section, indicating that there is energy loss in the transmission of vibration energy.

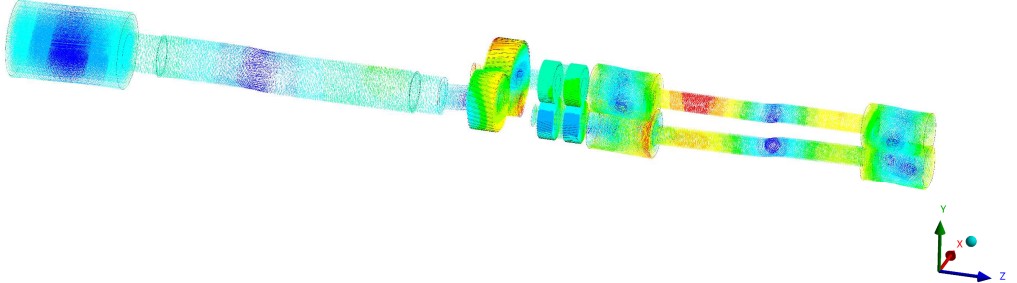

**Figure 6.** Vector diagram for vibration energy flow of 21 Hz.

The maximum value of the vibration energy flow under excitation of 43 Hz occurs on the upper joint drive side with an amplitude of 863.78 kW/m² (Figure 7). While the lower universal shaft joint shaft and motor in the indirect shaft on the vibration energy flow are smaller.

The energy flow vector on the mill drive system at 43 Hz gradually decreases from both ends of the upper knuckle shaft to both sides (Figure 8). There is an energy sink on the main drive system reducer pinion.

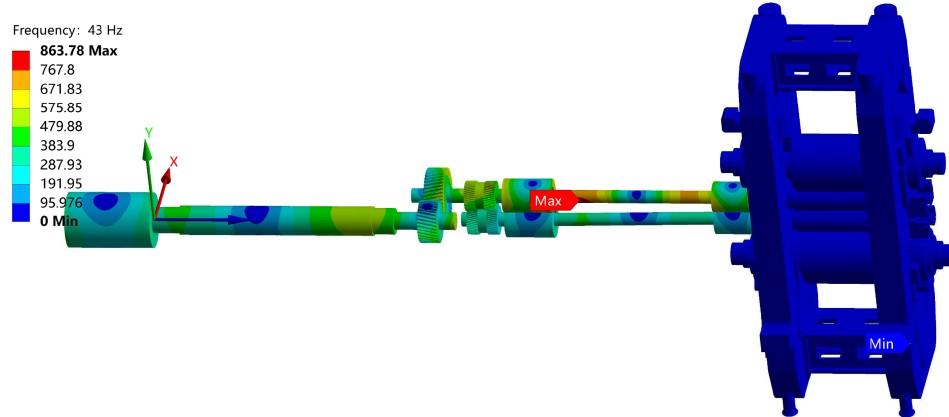

**Figure 7.** Harmonic response cloud for vibration energy flow of 43 Hz.

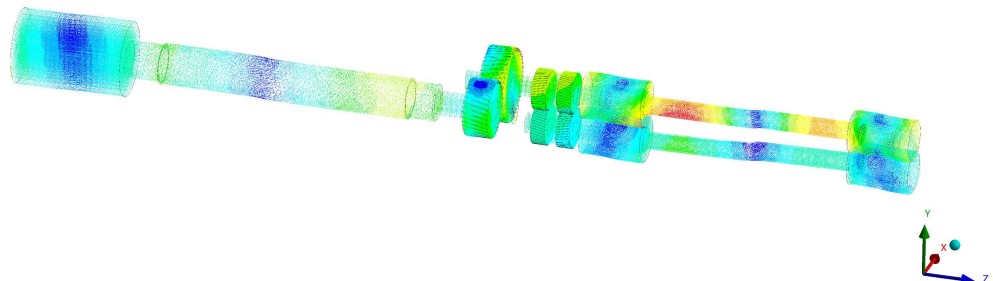

**Figure 8.** Vector diagram for vibration energy flow of 43 Hz.

The maximum value of vibration energy flow under excitation of 64 Hz appears on the operating side of the top of the mill deck with an amplitude of 887.35 kW/m$^2$ (Figure 9). The upper work roller operating side bearing seat vibration energy is large, and the amplitude is about 828 kW/m$^2$. The energy flow on the upper and lower work rolls is asymmetric, and the upper work roll vibration energy flow is a bit larger. Meanwhile, the upper support roll energy flow is also larger than the lower support roll energy flow.

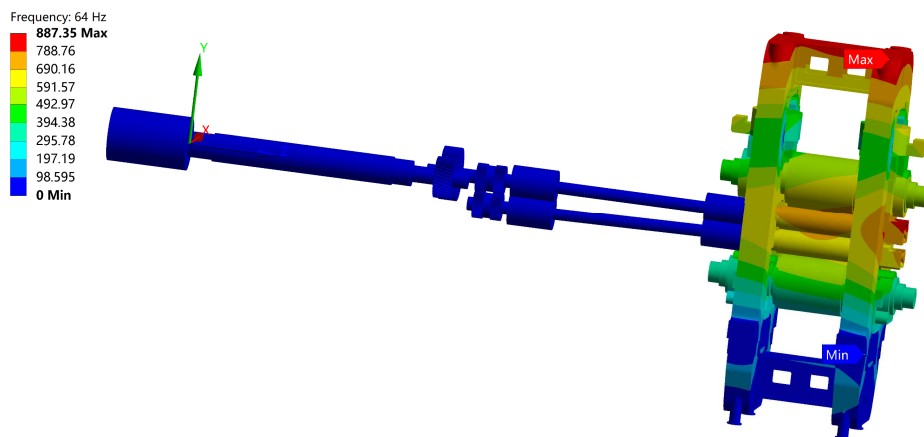

**Figure 9.** Harmonic response cloud for vibration energy flow of 64 Hz.

The energy flow vector of the vertical system flows from the upper and lower working rolls at the contact arc along the pagoda column to the top of the pagoda and gradually increases when the mill is excited at 64 Hz (Figure 10). There is very little energy flowing down to the lower support rolls, which may be related to the structure of the hot strip mill.

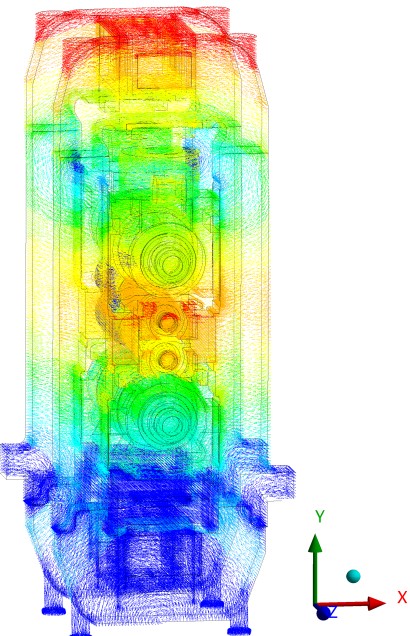

**Figure 10.** Vector diagram for vibration energy flow of 64 Hz.

The maximum value of vibration energy flow under an excitation of 121 Hz appears on the side of the pagoda column rolling direction with an amplitude of 827.52 kW/m$^2$ (Figure 11). The vibration energy of the upper working roller contact arc part is larger, and the vibration energy flow of the lower working roller and upper and lower support roller surface is smaller, which is asymmetric.

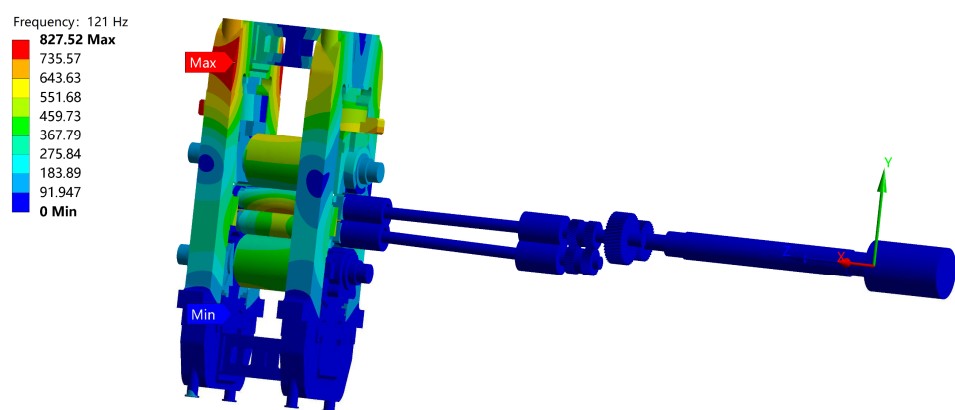

**Figure 11.** Harmonic response cloud for vibration energy flow of 121 Hz.

The energy flow vector of the vertical system flows from the contact arc of the upper and lower work rollers along the upper support rollers to both sides of the pagoda column, and it gradually increases (Figure 12).

### 4.2. Effect of Moment Dynamic Load on Vibration Energy of Rolling Mill Stand

The fluctuation of the preload torque has no effect on the magnitude of the vibration energy flow of the main drive system of the mill (Figure 13a). It shows that the torque fluctuation changes linearly with the magnitude of the vibration energy flow in the main drive system of the mill (Figure 13b). It can be considered that when the rolling torque is a certain stable value, the vibration energy does not increase, while, when frequent and small fluctuations of the rolling torque occur to excite the accumulation of vibration energy to form self-excited vibration, the energy flow rapidly becomes large.

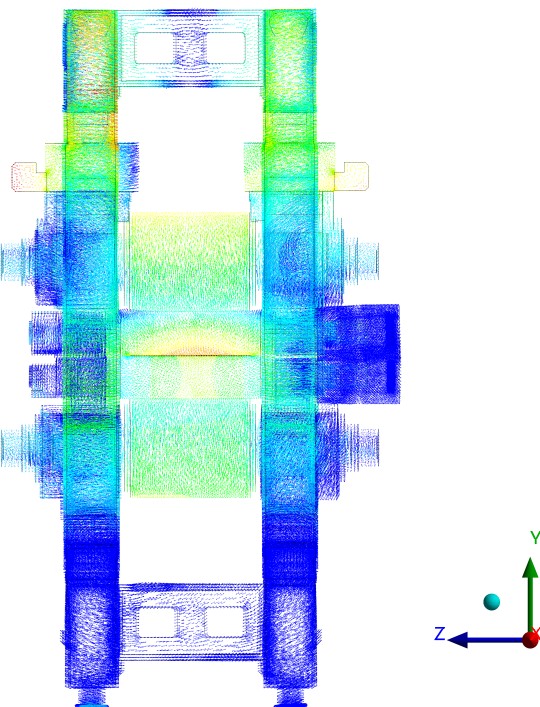

**Figure 12.** Vector diagram for vibration energy flow of 121 Hz.

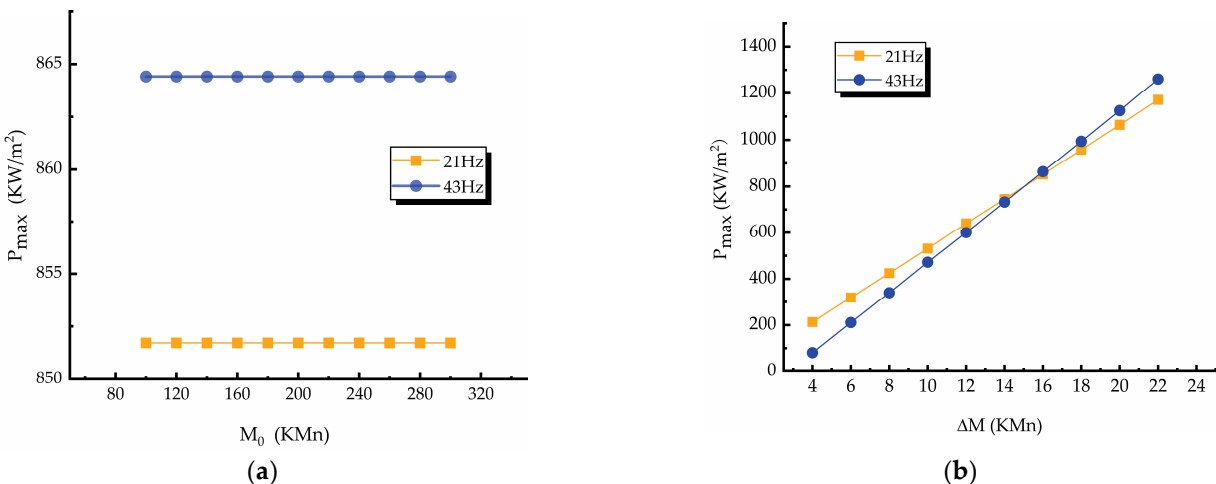

**Figure 13.** Moment dynamic load on impact of rolling mill vibration energy: (**a**) effect of preload torque variation on energy flow and (**b**) effect of torque fluctuations on energy flow.

Fluctuations in the steady-state rolling forces have no effect on the magnitude of the vibration energy flow in the vertical system of the mill (Figure 14a). It shows that the fluctuation of the rolling force changes linearly with the magnitude of the vibration energy flow in the vertical system of the mill (Figure 14b). We believe that, in the rolling process, frequent and small fluctuations in the rolling force caused the accumulation of mill vibration energy to form self-excited vibration, prompting an accelerated increase in the vibration energy flow.

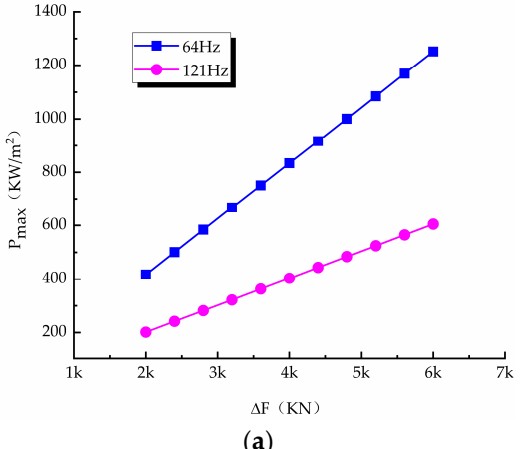
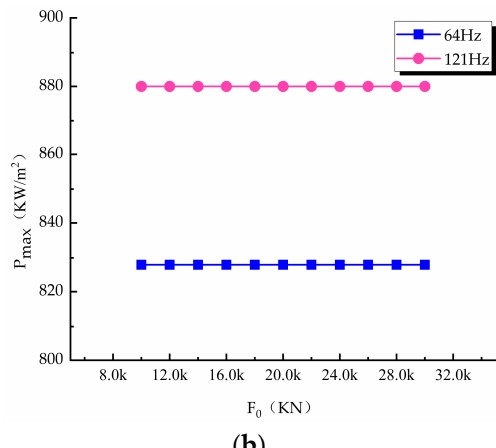

(**a**)      (**b**)

**Figure 14.** Rolling force dynamic load on impact of rolling mill vibration energy: (**a**) steady-state rolling forces on impact of mill vibration energy and (**b**) rolling force fluctuations on impact of rolling mill vibration energy.

### 4.3. Effect of Intersection Difference of Double Disturbance Load on Vibration Energy

As shown in Figure 15, the energy flow of mill vibration under excitations of 21 Hz, 43 Hz, 64 Hz, and 121 Hz with the phase angle difference shows a "V" change, the minimum energy value for excitations of 21 Hz and 43 Hz appears in the rolling force fluctuations and torque fluctuations with a phase angle difference of 180°, and the minimum energy value for excitations of 64 Hz and 121 Hz appears in the rolling force fluctuations and torque fluctuations with a phase angle difference of 200°. The minimum energy value corresponding to the 20° deviation of the phase angle in the vertical torsional coupling may be caused by the different material models and different settings of the structural constraints of the components.

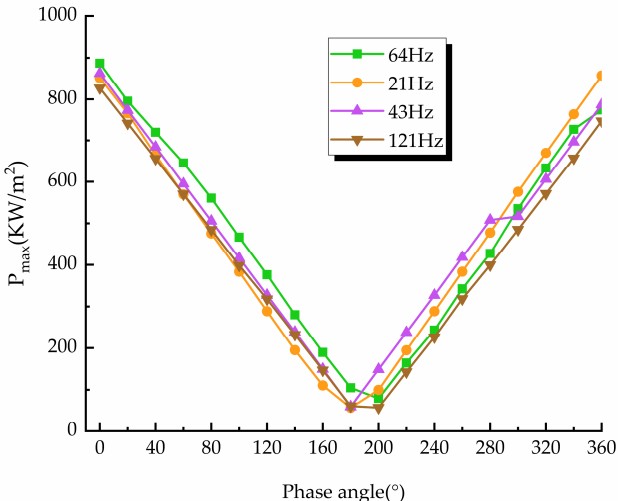

**Figure 15.** Effect of dynamic load intersection difference on vibration energy flow.

### 4.4. Effect of Damping Ratio on Vibration Energy

The maximum vibration energy response of the mill under different damping ratios is shown in Figure 16. The vibration energy reaches the maximum magnitude at each vibration frequency when the damping ratio is 0.001. The vibration energy at 21 Hz and 64 Hz is significantly reduced when the damping ratio is increased by a factor of 10 to 0.01, and the amplitude is reduced by 20%. However, this damping amount has a very small effect at 43 Hz and 121 Hz. When the damping ratio is increased by 100 times to 0.1,

the vibration energy of all resonant frequencies is significantly reduced by approximately 86%. When the damping ratio reaches the critical damping, that is, it increases to 1.0, the vibration energy has no amplitude.

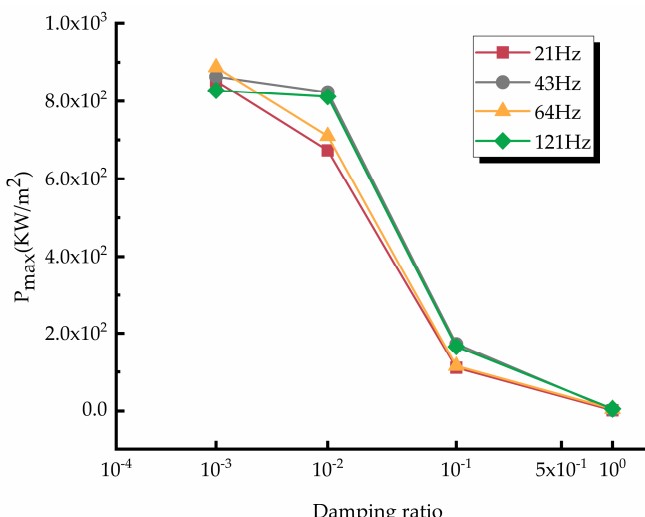

**Figure 16.** Effect of damping ratio on maximum vibration energy.

Increasing the contact arc area between the upper and lower work rolls of the mill and strip can effectively increase the damping effect. A damping ratio of 0.01–0.1 is actually effective.

### 4.5. Effect of Strip Width on Vibration Energy

In our mill energy flow harmonic response analysis, we set the strip thickness to 10 mm and simulated every 100 mm strip width increase to obtain the dot product of vibration velocity harmonic response and stress harmonic response for each sensitive frequency. In the field, we also found, by the strip width meter, that the vibration traces on the surface of the strip weakened after the width increase. As shown in Figure 17, the strip width of the MCCR finishing mill has little effect on the vibration energy of the main drive system; however, it has a greater effect on the vibration energy of the vertical system, and the vibration energy amplitude decreases significantly as the bandwidth increases. This may be due to the wide strip having a better damping effect.

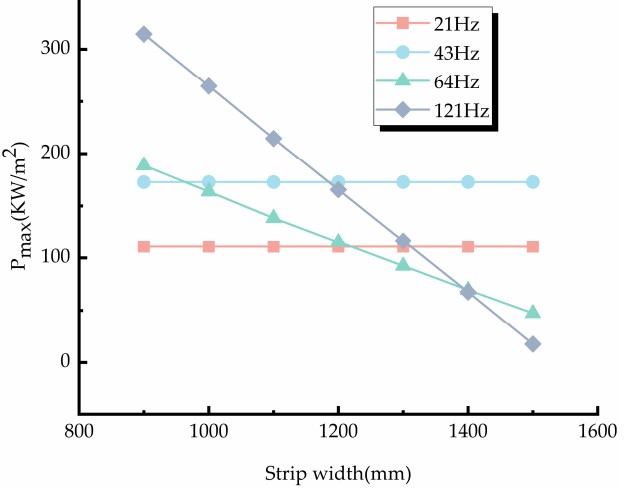

**Figure 17.** Effect of strip width on maximum vibration energy.

### 4.6. Effect of Strip Modulus on Vibration Energy

As shown in Figure 18, a small strip modulus corresponds to larger energy amplitudes of dip (64 Hz and 121 Hz). The opposite is true for the vibration of the main drive system, where the vibration energy at 21 Hz and 43 Hz increases significantly with increasing strip modulus. This result may be due to the fact that, for a given low modulus strip, the main drive system is more damped, while for the vertical system, the damping becomes less. Whether this law has a similar variation law for several other stands of the MCCR finishing rolling unit will be further investigated in the subsequent work.

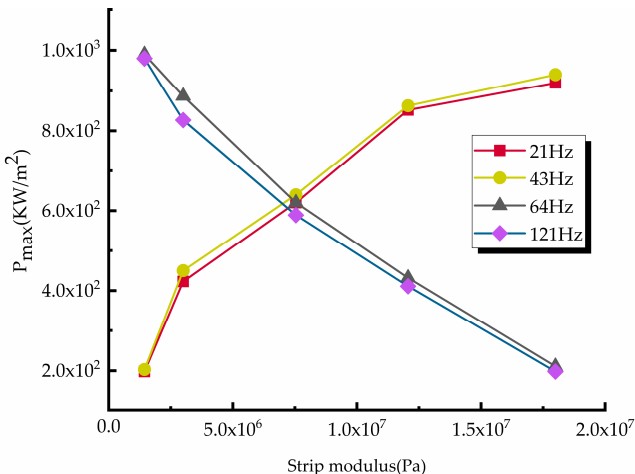

**Figure 18.** Effect of strip modulus on maximum vibration energy.

### 5. Conclusions

In this study, after analyzing the energy flow pattern and the energy flow harmonic response cloud of the mill, the following conclusions were drawn:

(1) When rolling SMCC slabs with a strip thickness of 1.5 mm and bandwidth of 1280 mm, the sensitive resonance frequencies of the mill drape-torsion coupling system were 21 Hz, 43 Hz, 64 Hz, and 121 Hz. These correspond to the 6th-, 10th-, 16th-, and 24th-order mode of the energy flow, respectively;

(2) The steady-state rolling force and preload torque changes have no effect on the amplitude of vibration energy flow. However, the rolling force fluctuation and torque fluctuation have a significant effect on the vibration energy flow. The vibration energy flow increases with an increase in the fluctuation. The phase angle difference between rolling torque shows a "V" trend on the vibration energy flow. The minimum vibration energy flow of the main drive system occurs at a phase angle difference of 180°, and the vertical system vibration energy flow occurs at a phase angle difference of 200°;

(3) When the damping ratio of the mill is 0.001–0.01, the energy flow reduces significantly under excitations of 21 Hz and 64 Hz but reduces very little under excitations of 43 Hz and 121 Hz. When the damping ratio is 0.01–0.1, the reduction of the vibration energy flow under all excitation frequencies reaches 86%, and the vibration suppression effect is obvious;

(4) The strip width variation has a large effect on the vibration energy flow of the vertical system but has little effect on the main drive system;

(5) An increase in the strip modulus causes an increase in the vibration energy flow in the main drive system but reduces the vibration energy flow in the vertical system. Whether this trend is also consistent with the F1, F2, F4, and F5 frames will be further investigated in subsequent work.

**Author Contributions:** Conceptualization, L.W.; methodology, L.W. and X.Y.; software, L.W.; validation, L.W., S.W., and X.J.; formal analysis, X.W.; investigation, X.J.; data curation, X.Y. and S.W; writing—original draft preparation, L.W.; writing—review and editing, L.W.; visualization, L.W.;

supervision, X.Y.; project administration, X.Y.; funding acquisition, X.Y. All authors have read and agreed to the published version of the manuscript.

**Funding:** This research is supported by the Fundamental Research Fund Project of Central Universities (Grant No. FRF-AT-19-001).

**Institutional Review Board Statement:** Not applicable.

**Informed Consent Statement:** Not applicable.

**Data Availability Statement:** Not applicable.

**Conflicts of Interest:** The authors declare no conflict of interest.

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
