# Peer review of "Vibration Energy Coupling Behavior of Rolling Mills under Double Disturbance Conditions"

_electronics, doi:10.3390/electronics12041061_

Round 1

Reviewer 1 Report

The article investigated the mill vibration energy flow modal and vibration energy flow harmonic responses of multimode continuous casting and rolling plant F3 (MCCR) by building an overall finite element model. So, the factors affecting the mill vibration energy flow are analyzed further. This is a detailed and comprehensive article whose sections are arranged logically and reasonably.

The authors lead the reader logically to the purpose of the paper with a general introduction to mill vibration. Then, mechanisms of the vibration and vibration control of rolling mills are outlined, along with related studies of these that could not fully explain the long-term mill vibrations. Under double disturbance conditions, the authors can get the results of the mill vibration energy flow calculations and their influencing factors through numerical calculations.

Following the introduction, the authors' calculation using the sound intensity method is presented in the article. The structural sound intensity (eq1) and the mill model are shown in Figure 1. The vibration power flow in the frequency domain for the cylinder elements, shell elements, and beam elements can be expressed by the Fourier transform, which is indicated in equations 3, 4, and 5. Moreover, the finite element modeling of the coupled structure of the mill is illustrated in figure 2, and its structural parameters are shown in table 1. From all the methods above, harmonic response analysis is used to find the vibration velocity, and the authors conclude the important factors that need to be clarified in the article, such as frequencies and frequent vibrations. The next part is also the most important, the authors present the results obtained from the simulation models at frequent vibrations of 21, 43, 64, and 121 Hz. The effect of moment dynamics, intersection difference of double disturbance, damping ratio, strip width, and strip modulus on vibration energy is processed and aggregated in Figures 13 to 16.

In the conclusion, the authors have pointed out 5 important points that they obtained from the research. This article is a very complete and careful explanation of the factors affecting mill vibration energy flow to optimize the performance of multimode continuous casting and rolling plant F3 (MCCR). There are two points that authors may consider when editing, which are:

First, adding some explanation to Section 4, for example, in lines 233-234, what factors make vibration energy at 21 Hz and 43 Hz increase significantly with increasing strip modulus?

Second, the conclusion, lines 240–244, is about the tools used in the study; this paragraph can be moved up to the introduction so that the reader is aware of these tools from the start, allowing the conclusion to focus solely on the research results. The softwares and the codes used in this study should be in details or as a supplementation (if codes are used).

And others:

In the Abstract section and the following sections, the author should explain about "F3" term, what is it?

Also in the Abstract section, the author should explain more clearly about the results of this research so that readers can better understand the topic as well as everything written in the article.

How the torque stability is measured with what type of device, as the result shown in Figure 3? Please show it's specification.

Before each equation, the author should explain why there is that equation, what it is so that the reader can clearly understand the equation.

What is SMCC. As shown in Table 1, SMCC is a kind of steel grade?

Author Response

Please see the attchment

Reviewer 2 Report

A very interesting article, however, it needs to be supplemented with important information. The resonant frequencies depend on: the type of material, the shape of the part and the method of support. The authors did not provide information on the material models used and did not provide full information on how the degrees of freedom were obtained. From the industrial point of view, it is also important to indicate the existing technological defects of the process and to prove that the introduced changes improve the technological quality of the manufactured parts. Introducing the proposed changes will improve the article submitted for review.

Reviewer 3 Report

Paper seems to be Scientific Soundness. The following points to be addressed,

1. Provide more points on structural sound intensity method and harmonic response analysis  method

2. Add few more literatures.

3. What is the full form of SMCC and MCCR?

4. How authors have confirmed that The strip width variation has a large effect on the vibration energy flow of the vertical system?

5. Elaborate the Novelty of the research paper.

Round 2

Reviewer 1 Report

The revised version of round 2 is well prepared and significantly improved.

Only, the SMCC is not steel grade. It should conform to the Chinese standard, such as GB/T 283-2021, or another equivalent.
